# Detection of Selected Canine Viruses in Nigerian Free-Ranging Dogs Traded for Meat Consumption

**DOI:** 10.3390/ani13061119

**Published:** 2023-03-22

**Authors:** Linda A. Ndiana, Gianvito Lanave, Costantina Desario, Amienwanlen E. Odigie, Kelechi G. Madubuike, Maria Stella Lucente, Chukwuemeka A. Ezeifeka, Giovanni Patruno, Eleonora Lorusso, Gabriella Elia, Canio Buonavoglia, Nicola Decaro

**Affiliations:** 1Department of Veterinary Medicine, University of Bari, Strada Provinciale per Casamassima Km 3, Valenzano, 70010 Bari, Italyamienwanlen.odigie@uniba.it (A.E.O.);; 2Department of Veterinary Microbiology, College of Veterinary Medicine, Michael Okpara University of Agriculture, Umuhaia Ikot Ekpene Road, Umudike 440101, Nigeria

**Keywords:** Nigeria, dog trade, selected viruses, molecular characterization, genetic mutations

## Abstract

**Simple Summary:**

Dog meat is a delicacy in many countries of the world, including Nigeria. Here, we report the identification of selected canine viruses in trade dogs. The results showed that canine parvovirus circulates in trade dogs at a high frequency and provides the first molecular evidence of canine distemper virus (CDV) and canine circovirus (CaCV) in that country.

**Abstract:**

Animal trade favors the spreading of emerging and re-emerging pathogens. Concerns have been previously expressed regarding the risks of dog trade in spreading zoonotic pathogens in Nigeria. However, the role of these dogs in disseminating highly pathogenic canine viruses has not yet been explored. The present study aimed to identify selected canine viruses in dogs traded for meat consumption in Nigeria. A total of 100 blood samples were screened for carnivore protoparvovirus-1 (CPPV-1), canine adenovirus 1/2 (CAdV-1/2), canine circovirus (CaCV), and canine distemper virus (CDV) by using real-time PCR and conventional PCR and/or sequencing. CPPV-1 DNA was identified in 83% of canine samples while CaCV DNA and CDV RNA were detected in 14% and 17% of the dog samples, respectively. None of the dogs tested positive for CAdV-1/2. The CaCVs identified in this study clustered along with other European, Asian, and American strains. Moreover, CDV strains identified in Nigeria clustered in a separate lineage with the closest genetic relatedness to the Europe–South America-1 clade. Further surveys prior to and after arrival of dogs at the slaughtering points are required to clarify the real virus burden in these animals.

## 1. Introduction

Through history, viruses have posed a huge threat to domestic dogs globally. These include canine parvovirus 2 (CPV-2), canine distemper virus (CDV), canine adenovirus 1/2 (CAdV-1/2), and the relatively new canine circovirus (CaCV). CPV became widespread within few years of its initial report in the late 1970s. The introduction of specific vaccines aided its control, but the virus is still far from being eradicated [1,2].

CPV-2 causes mortality especially in puppies with waned or absent maternal/vaccine-derived virus-specific antibody [3]. CPV-2 is a member of the species Carnivore protoparvovirus 1 (CPPV-1, genus *Protoparvovirus*, family *Parvoviridae*) [4], with a genome approximately 5 kb long. This consists of two major ORFs, NS (NS1 and NS2) and VP (VP1 and VP2), encoding the nonstructural and structural proteins, respectively [5]. VP2 makes up about 90% of the viral capsid proteins and bears the antibody-inducing epitopes, thereby eliciting the immune response [6]. This protein is hypervariable and is typically used as the basis for the characterization of CPV. There are now three main variants of the first recognized CPV-2—2a, 2b, and 2c—which all emerged within a decade of the initial report and shortly replaced the original virus [7,8]. Since the first report of CPV in Nigeria in 1985 [9], a dynamic epidemiology of the virus has been observed, with CPV-2a being first identified in 2013 [10], and all three variants were subsequently found to be circulating in domestic dogs in the country [11]. However, CPV-2c was retrospectively found to have been circulating in the country at least since 2016 [12]. The latest findings suggest that the CPV-2c strain, which was found in dogs in 2016, is the currently circulating predominant virus type and appears to be of Asian origin due to its similarity to previously reported viruses from that continent. Possible modes of introduction and spread of this new variant, however, could not be ascertained.

CaCV is a relatively newly discovered member of the genus *Circovirus* along with *Cyclovirus*, in the family *Circoviridae* [13]. CaCV is a DNA virus with a circular, ambisense, single-stranded genome enclosed in a nonenveloped icosahedral capsid [14]. CaCV has been widely reported in domestic dogs and is associated either with enteric [15,16,17,18], respiratory [19,20] or systemic [21,22] diseases. The pathogenesis of this virus is not yet well understood although it has been suggested to possibly suppress the host immunity [22].

CAdVs belong to the genus *Mastadenovirus* in the family *Adenoviridae* [23]. They have a double-stranded DNA genome about 32 kb long. A CAdV associated with infectious canine hepatitis (ICH) is known as CAdV-1, while another genetically related virus which causes a mild respiratory disease (kennel cough) is referred to as CAdV-2 [24]. Vaccination against CAdV-1 is widely practiced and has controlled the spread of the virus in developed countries; however, the virus persists and occasionally re-emerges in dog populations worldwide.

CDV belongs to the species Canine morbillivirus under the genus *Morbillivirus* in the family *Paramyxoviridae* and is an enveloped RNA virus with nonsegmented, negative-sense, single-stranded genome [25]. The virus is transmitted by aerosol and has multicell tropism, causing canine distemper, a severe and often fatal disease involving the respiratory, digestive, urinary, lymphatic, cutaneous, skeletal, and central nervous systems [26]. The virus has been reported in multiple carnivore species including aquatic animals. The CDV genome is about 15 kb long with 6 ORFs (N-P-M-F-H-L) separated by intergenic untranslated regions. These genes give rise to 8 respective proteins: nucleocapsid protein, phosphoprotein (P), matrix protein, fusion protein, hemagglutinin, and large polymerase. They also encode the C and V proteins, which are alternative gene expressions of the P gene [27,28]. Hemagglutinin, encoded by the hypervariable H gene, is the transmembrane protein involved in virus–receptor interaction, thereby influencing the antigenicity, cell tropism, host range, and pathogenicity of CDV. Therefore, the H gene has been the focus of research into CDV variability and evolution [28].

Clinical and serological evidence of CDV has been obtained among Nigerian domestic dogs [29], while no molecular evidence of either CDV, CAdV-1/2, or CaCV has been reported in Nigeria. Nigeria is one of the countries in the world today where dogs are vastly traded for human consumption. Huge dog markets exist in the northern part of the country [30,31], from where free-ranging indigenous breeds are transported in stacked communal cages, by road, to various parts of the country. Different tribes consume dogs for various reasons, including medicinal purposes [32]. However, in the southern part of Nigeria, dog meat is a relished delicacy [33].

Over the years, there have been major concerns regarding the risk of human exposure to deadly pathogens, such as rabies, possibly harbored by these dogs [31,33]. However, these animals may also impact the epidemiology of other important viruses of carnivores in the country. In the present study, we investigated the presence of CPV, CDV, CAdV-1/2, and CaCV in dogs slaughtered for meat production in Nigeria.

## 2. Materials and Methods

### 2.1. Sample Collection and Nucleic Acid Extraction

The study area was Uyo, Akwa-Ibom state (5.0377° N, 7.9128° E), a 140-square-mile city in southern Nigeria. Shape files of study locations were obtained from the ArcGIS online map tools and imported for visualization into ArcGIS version 10.8.1 (Redlands, CA, USA: Environmental Systems Research Institute, Inc., 2020). Sampling was carried out at three retail/slaughter points at which about 7 dogs were displayed weekly for purchase by the final consumers, most of whom retrieved their animals live, while others demanded prior slaughter. These were the major middlemen involved in the supply of dog-meat consumption in Uyo, the capital city of Akwa Ibom state, and the slaughter slabs were designated points 1, 2, and 3 (Figure 1). All the dogs on display at these sampling points appeared dehydrated and low in body weight, probably due to prolonged starvation during the journey from the northern part of the country and during holding, although they showed no obvious signs of disease. Blood was collected with convenience sampling from 100 dogs slaughtered between August 2020 and March 2021, with an average of 3 samples collected weekly. 

The samples were stored at −20 °C at the collection points, after which they were transported under cold chain to the Infectious Diseases Unit, Department of Veterinary Medicine, University of Bari, and stored at −80 °C until analyses. Viral nucleic acid was extracted from 100 μL of blood sample, diluted in an equal volume of PBS using the IndiSpin Pathogen Kit (Indical Bioscience GmbH, Leipzig, Germany) according to the manufacturer’s instruction, and stored at −20 °C. Ethical approval for the study was obtained from the College of Veterinary Medicine ethics committee of Michael Okpara University of Agriculture, Umudike, with approval number MOUAU/CVM/REC/202118.

### 2.2. Screening for Canine Viruses

Screening of extracts for the presence of viral nucleic acid was performed with real-time PCR (qPCR) assays. A TaqMan-based qPCR protocol was used for the initial screening for the presence of CPV DNA [34]. CPV-positive samples were further characterized by minor groove binder (MGB) probe-based qPCR assays to differentiate CPV types 2a/2b and 2b/2c as well as vaccine CPV-2 strains from field variants [35] (Appendix A).

Extracts were also screened for the presence of CaCV DNA using specific primers and probe targeting of the rep gene of CaCV [14]. CAdV DNA was detected using an qPCR protocol with a primer pair and probes that discriminate between CAdV-1 and CAdV-2 [36] (Appendix A).

Reverse transcription was performed on extracts to a total reaction volume of 20 μL containing PCR buffer 1×, 5 mM of MgCl_2_, 1 mM of deoxynucleotide, RNase Inhibitor 1 U, 2.5 U of reverse transcriptase, and 2.5 U of random hexamers. Cycling conditions were as follows: 42 °C for 30 min, followed by a denaturation step at 99 °C for 5 min. Reverse transcripts were then screened for the presence of CDV RNA [37]. 

Real-time PCRs were performed using iTaq™ Universal Probes Supermix (Bio-Rad Laboratories SRL, Segrate, Italy).

### 2.3. PCR Amplification, Sequencing, and Sequence Analyses

Two overlapping fragments of the CPPV-1 VP2 gene (742 bp and 1375 bp) were amplified using previously designed primer sets [38,39] (Appendix A). PCR amplification of a 400 bp fragment of the rep gene of circoviruses was carried out with a nested PCR using broadly reactive primers [40] (Appendix A). Reverse transcription (RT)-PCR was performed with primer pair H5F and P5 [41,42], which is able to amplify a 770 bp fragment of the partial H and L genes of CDV separated by the intergenic region (Appendix A).

All PCR assays were performed in a final volume of 50 μL and contained 5 μL of DNA extract, 24.5 μL of PCR grade water, TaKaRa LA TaqTM Kit (Takara Bio Europe S.A.S. Saint-Germain-en-Laye, France) consisting of 5 μL of 10x buffer, 5 μL of MgCl^2^ (25 mM), 1 μL of forward and reverse primers (50 μM), and 8 μL of deoxynucleotides. Thermal cycling conditions for CPV amplification included an initial denaturation at 94 °C for 2 min, 35 cycles of denaturation at 94 °C for 30 s, annealing at 58 °C for 30 s. and extension at 68 °C for 1 min, followed by a final extension at 68 °C for 10 min. The cycling conditions for the primary and nested PCRs for the amplification of the CaCV DNA included an initial denaturation at 94 °C for 2 min, 35 cycles of denaturation at 94 °C for 15 s, annealing at 52 °C for 15 s, and extension at 68 °C for 15 s, followed by a final extension at 72 for 10 min. The cycling conditions for CDV included an initial denaturation at 94 °C for 2 min, 35 cycles of denaturation at 94 °C for 15 s, annealing at 52 °C for 30 s, and extension at 68 °C for 15 s, followed by a final extension at 68 for 10 min.

PCR products were purified using the Qiaquick PCR purification Kit (Qiagen GmbH, Hilden, Germany) and subsequently sequenced in both directions using BigDye 3.1 Ready Reaction Mix (Applied Biosystems) according to the manufacturer’s instructions. The genomic sequences obtained in this study were assembled and analyzed using Geneious Prime software package version 2021.2 (Biomatters Ltd., Auckland, New Zealand).

The web-based tools Basic Local Alignment Search Tool (BLAST; http://www.ncbi.nlm.nih.gov, accessed on 17 January 2023) and FASTA (http://www.ebi.ac.uk/fasta33, accessed on 17 January 2023) were employed using the default values to find homologous hits. Sequence editing was carried out using Geneious Prime version 2021.2 (Biomatters Ltd., Auckland, New Zealand).

The obtained CPV, CDV, and CaCV strain sequences were separately aligned with cognate strains retrieved from the GenBank database with multiple alignment using the Fast Fourier Transform (MAFFT) algorithm [43]. The appropriate substitution model was assessed using “Find the best protein DNA/Protein Models” of MEGA X version 10.0.5 software [44].

Phylogenetic analyses and evaluation of selection pressure on coding sequences were obtained for CPV sequences using the maximum-likelihood method, Tamura-Nei 4-parameter model, and a discrete gamma distribution with 6 categories and 1000 replicates as statistical support were used. The maximum-likelihood method, Kimura 2-parameter model, and a discrete gamma distribution and a proportion of invariant sites including 6 categories with 1000 replicates as statistical support were used for CaCV strains. The maximum-likelihood method, Tamura-Nei 4-parameter model, and a discrete gamma distribution with 6 categories and 1000 replicates as statistical support were applied for CDV strains. Bayesian inference and neighbor joining methods for the phylogeny were also explored, exhibiting similar topologies; the maximum-likelihood tree was finally kept.

## 3. Results

Blood samples collected from 100 dogs tested positive under qPCR to CPPV-1 DNA (83/100, 83%), displaying cycle threshold (CT) values ranging from 15 to 34 (mean CT value = 33; median CT value = 34) (Table 1). A total of 35 CPPV-1-positive samples could be characterized with qPCR. CPV-2c was identified in 77.1% (27/35) of the characterized samples and in coinfections, with CPV-2a in 8.6% (3/35) of the samples. CPV-2a was also detected in the remaining 14.3% (5/35) of dog samples. No CPV-2b was identified in this study.

Due to the low virus titers of the specimens, only twenty-two CPV-2-positive samples could be successfully amplified in the full VP2 gene using conventional PCR and sequenced. The CPV-2 strains identified shared 98.2–100% nucleotide (nt) identities to cognate CPV-2 strains according to BlastN and FASTA analysis. The phylogenetic tree appeared polyphyletic, with CPV-2c viruses clustering into three distinct clades (I, II, and V), CPV-2a viruses into three different clades (III, IV, and VIII), and CPV-2b viruses into two different clades (VI and VII). Twenty-one strains identified in this study clustered with other Nigerian [45,46], Asian, and European CPV-2c strains. Strain NGA/2021/265.21-79 (GenBank accession no. ON063556) segregated together with CPV-2a strains detected in Nigeria [45,46], China, Thailand, India, the USA, and Uruguay (Figure 2).

In total, twenty-one and thirty-six nt mutations were observed in CPV-2c and CPV-2a sequences, respectively. Deduced VP2 aa sequences were compared with the analogous sequences of CPV-2a and CPV-2c strains previously detected in Nigeria [45,46] (Appendix A). Fifteen and one nonsynonymous mutations were observed in the identified CPV2c and CPV-2a sequences, respectively. All the CPV-2c strains displayed residues Gly at position 5 and Arg at position 370, similar to other Nigerian strains [45,46] (Appendix A). The remaining thirteen nonsynonymous mutations in the CPV-2c sequences were unique, ten of which have never been reported before (Ser226Gly, Pro229Ala, Gln350His, Arg382Thr, Phe499Ile, Arg520Gly, Gln558Pro, Asn560Thr, Lys582Asn, and Tyr584Phe). The other three nonsynonymous mutations, Gln365His, Val424Ala, and Asn565Lys, have been previously described elsewhere [10,47,48].

The single nonsynonymous mutation for the CPV-2a sequence NGA/2021/265.21-76 (ON0635455) consisted of Thr440Ala (Appendix A), formerly observed in Nigeria [10,11,45,46].

CaCV DNA (14/100, 14%) was identified using qPCR with CT values varying from 28 to 35 (mean CT value = 33; median CT value = 35) (Table 1). Partial rep gene sequences could be obtained from only six CaCV-positive samples due to the low viral DNA content of most samples. CaCV strains identified in this study shared a 92.0–98.5% nt relatedness with other CaCV sequences according to BlastN and FASTA analysis. In the phylogenetic tree based on the partial nucleotide sequence of the rep gene, the CaCV strains identified in this study clustered in clade 1 along with other European, Asian, and American CaCV strains (Figure 3).

CDV RNA (17/100, 17%) was also retrieved from Nigerian samples with CT values between 22 and 40 (mean CT value = 33; median CT value = 33) (Table 1). Partial H and L gene sequences could be amplified from 2 Nigerian CDV-positive samples. The two sequences shared 91.71–96.99% nt identity with other CDV strains in the GenBank database according to BlastN and FASTA analysis. In the phylogenetic tree based on the partial nucleotide sequence of the hemagglutinin gene, the CDV strains identified in this study were basal to the clade Europe–South America-1 (Figure 4). The deduced amino acid sequence of the partial H gene of the Nigerian strains revealed the presence of two nonsynonymous mutations (Gln580Arg and His549Tyr) compared to other CDV strains used for the phylogeny.

Furthermore, all blood samples from the Nigerian dogs tested negative for CAdV-1/2 DNA (Table 1).

## 4. Discussion

There have been efforts over the years to eradicate many viruses of carnivores such as CPV, CDV, and CAdV-1 without success [2]. While the high mutation rate and associated possible change in virus properties necessitates a continuous surveillance for CPV worldwide, an understanding of the geographical occurrence of the other important viruses of dogs is crucial in the implementation of virus control strategies. This report provides the first molecular evidence for the circulation of CaCV and CDV in Nigerian dogs and corroborates the results from a previous study that detected CPV-2c of possible Asian origin at a high frequency, as well as the absence of CPV-2b in this country [45]. 

The rate of detection of CPV in the tested dogs (83%) was nearly double that previously described in Nigerian domestic dogs. Former studies obtained detection rates for CPV ranging from 45 to 52% in dogs with clinical signs [12,45] and 17% in animals without clinical signs [49]. Considering the high infectivity of CPV, the high rate of positivity may be attributable to the lack of prophylaxis in the tested dogs. Furthermore, the close contacts at the dog markets, as well as in the course of their transportation, may predispose the healthy animals to acquire the infection. Although puppies are known to be relatively more susceptible to CPV infection than are adult dogs [7,49,50,51,52], a high rate of detection in these indigenous adult dogs supports recent reports accounting for their higher susceptibility to CPV [46] and for their potential role as reservoir hosts, harboring the infection without apparent clinical signs [49].

The CPV-2c strains analyzed in this study were nearly identical to the previously reported 2c strains that were suggested to have originated in Asia [12,45,53]. The earliest detection of this variant in Nigeria was in puppies sampled in 2016 [12]. Different from what is observed in wild carnivores with CPV seeming to evolve independently in the animals [54], this study demonstrates the circulation of identical viruses in domestic and free-ranging dogs, clearly suggesting transmission of the virus between these groups, although it is difficult to predict the specific point of infection, as an open dog market is a potential hub for transmission of pathogens. Considering the association of some of the recent mutations with the vaccine pressure, unvaccinated dogs may currently serve as vessels for CPV-2 amplification and dissemination across the country. Further surveys prior to and after the arrival of dogs at the slaughtering points are required to clarify the real virus burden in these animals.

A total of 22 CPPV-1 strains were characterized in this study. All the CPV-2c strains retrieved from the Nigerian dogs displayed the residue Gly at position 5 of the VP2 protein. The mutation Gly5Ala was also harbored by other Nigerian CPV-2c strains [45,46] and was first described in Chinese strains [55]. Moreover, all the CPV-2c strains identified in this report along with other Nigerian CPV-2c strains [45,46] exhibited the residue Arg at position 370. This unique Gln370Arg substitution was previously highlighted as evidence of a potential CPV-2c variant or new CPV-2c in Chinese [55,56] and Nigerian [12] dogs. Strain NGA/2021/265.21-59 (GenBank accession no. ON0635451) exhibited four novel mutations (Arg520Gly, Asn560Thr, Lys582Asn, and Tyr584Phe), while strain NGA/2021/265.21-11 (GenBank accession ON0635445) displayed another two mutations (Pro229Ala and Phe499Ile) that have not been previously identified. Strain NGA/2021/265.21-79 (ON0635456) exhibited two novel mutations (Gln350His and Arg382Thr) and an aa change (Gln365His) previously reported in a Korean dog [48]. Residue 365 is positioned along with residues 370 and 375 in the flexible loop that effects host range. Accordingly, this mutation may be responsible for host range extension [57]. Strain NGA/2021/265.21-13 (ON0635446) presented a novel mutation (Ser226Gly) and an aa change (Val424Ala) formerly described in dogs in South Africa [10] despite lacking a clear association with host range mutation. Strain NGA/2021/265.21-18 (ON0635447) showed the novel mutation Gln558Pro and the aa change Asn565Lys that was also found in a 300-Ala CPV mutant emerged in dog cells [47]. Position 565 is directly adjacent to a known host range mutation site (VP2 position 564) that differs between FPV and CPV [58,59,60].

Strain CPV-2a NGA/2021/265.21-76 (ON0635455) exhibited the residue Ala at position 440, mirroring that previously observed in most of the Nigerian CPV-2a strains [10,11,45,46]. The Thr440Ala substitution was formerly reported in Brazilian CPV-2a [61] and Chinese CPV-2b and CPV-2c strains [56,62]. Residue 440 is known to be an important factor for antigenicity [63], and the mutation to Ala along with other aa substitutions appears to be associated with selective pressure and the development of disease in vaccinated dogs [61].

CaCV-positive samples were identified with a 14% prevalence in Nigeria. Overall, since the first detection in 2012 [64], CaCV has been detected in dogs in different countries worldwide with a prevalence ranging from 1.34% to 19.8% [17,65,66,67,68,69]. However, no report of CaCV detection has been published to date in Africa, thus hinting at the current understanding regarding the circulation of this virus within and between African countries. In this study, no clinical data were available for the blood samples of the slaughtered dogs, thus limiting the assessment of any association with clinical presentation and the understanding of the epidemiology of CaCV in Nigeria. 

A total of six CaCV strains could be amplified in the partial rep gene sequence. Phylogenetic analysis provided evidence of CaCV clustering into six groups, as previously observed [65]. Nigerian strains clustered in group 1, as did the CaCV strains detected in Europe, America, and Asia, with no clear geographical pattern being observed.

CDV RNA was identified with a rate of positivity of 17% in the Nigerian dogs. CDV reached South Africa in the 1920s [70] and several decades later, a CDV epidemic involved wildlife of the Serengeti National Park (NP) in northeast Tanzania [71] spreading through the Serengeti ecosystem [72] and the Maasai Mara National Reserve in Kenya [73]. In canids, CDV infection was described in bat-eared foxes (latin name), and neurological signs were observed in silver-backed jackals (latin name) and golden jackals (latin name) [72]. Although CDV spillover from dogs can cause epidemics in wildlife, molecular and ecological studies have demonstrated that this virus can be maintained in wildlife populations [74,75]. For instance, a virulent CDV spread from dogs outside the Serengeti NP caused high mortality in lions inside the park [72]. In Nigeria, despite a lack of molecular evidence of CDV to date, some serological evidence has been reported [29]. 

The two strains detected In this study clustered apart from other CDV strains, thus representing a possible novel lineage. To date, twelve genetic lineages have been reported (America-1 and -2; Arctic-like; Asia-1, -2, and -3; Europe Wildlife; Europe-1/South America-1; South America-2; South America-3; Africa; and Rockborn-like) [76]. In Africa, the circulation of a single lineage has been reported: the Africa lineage includes CDV strains identified from a dog and other wild carnivore species [77].

Interestingly, both CDV sequences identified in this study retained Tyr and Arg in the hemagglutinin gene at positions 549 and 580, respectively. The CDV H gene determines viral tropism and host range, and an aa substitution from Arg to Gln at position 580 has been suggested to determine host jump to noncanid species [78]. However, both sequences from this study retained Arg at position 580. The aa Tyr was also retained at position 549, unlike some strains recently that have been reported in South America [79] and are regarded as part of the ancestral clade I, reported as early as 1980. 

The circulation of CDV in rural dogs is a recognized challenge [79,80], and the detection of CDV in Nigerian free-ranging dogs raises concerns regarding the risk posed by dog markets to pet dogs and other canids in their vicinity, as the average proximity of the sampling sites to the nearest forest reserve was only 10 km (Figure 1).

## 5. Conclusions

This study sheds light on the possible ongoing amplification and dissemination of some canine viruses in trade dogs in Nigeria and highlights the need to implement effective control measures to protect susceptible animals from infection.

## Figures and Tables

**Figure 1 animals-13-01119-f001:**
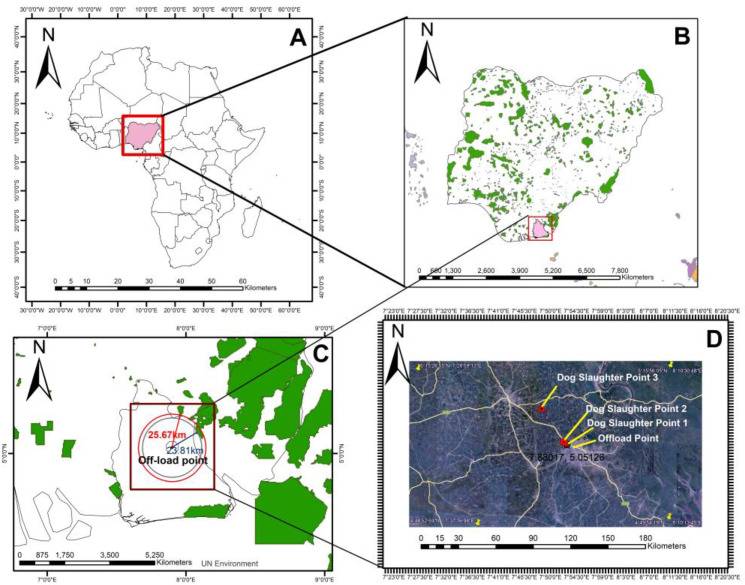
Map of the study area within the African continent (**A**), the country of Nigeria (**B**), and the Akwa Ibom state boundaries in Nigeria, (**C**) and the geographic distribution of dog slaughter slabs in Uyo, the capital city of Akwa Ibom state (**D**).

**Figure 2 animals-13-01119-f002:**
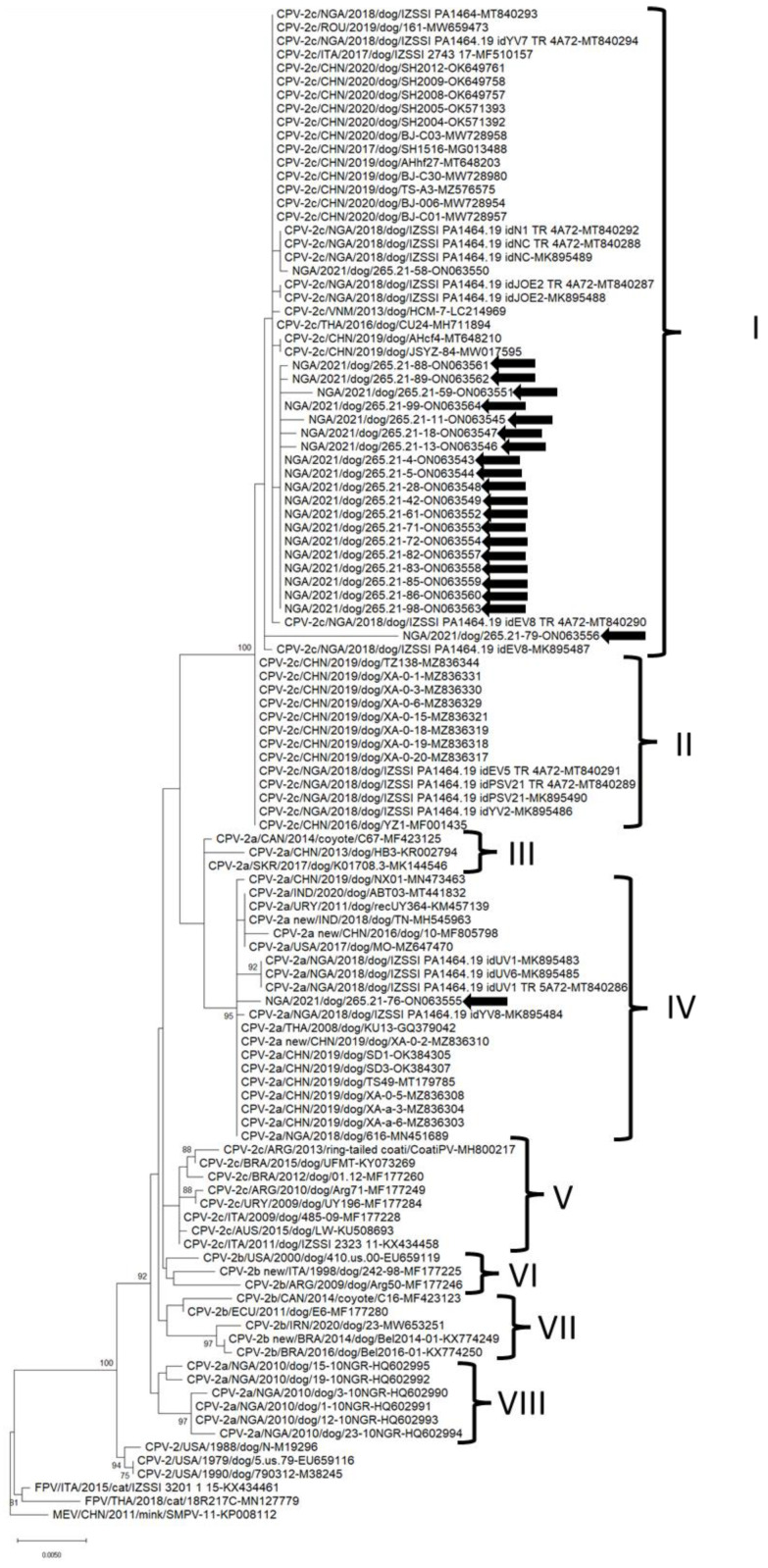
Phylogenetic tree based on the complete ORF2 sequence (1755 nt) of Carnivore Protoparvovirus-1 (CPPV-1). The maximum-likelihood method and Tamura-Nei model (four parameters) with a gamma distribution were used for the phylogeny. A total of 1000 bootstrap replicates were used to estimate the robustness of the individual nodes on the phylogenetic tree. Bootstrap values greater than 75% were indicated. The black arrows indicate CPPV-1 strains generated in this study and roman numerals indicate clades. The scale bar indicates the number of nt substitutions per site.

**Figure 3 animals-13-01119-f003:**
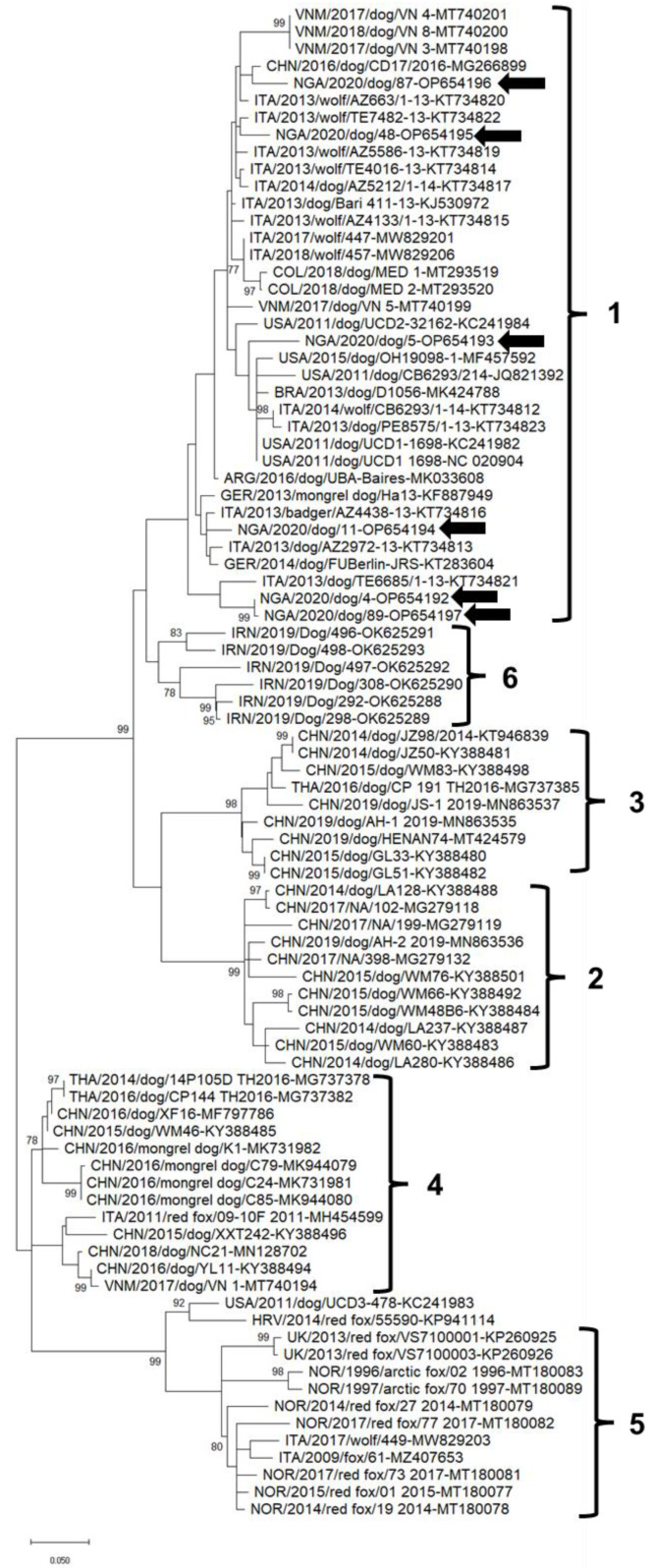
Phylogenetic tree based on the partial replicase sequence (307 nt) of dog circovirus (CV). The maximum-likelihood method and Kimura model (two parameters) with a gamma distribution and invariant sites were used for the phylogeny. A total of 1000 bootstrap replicates were used to estimate the robustness of the individual nodes on the phylogenetic tree. Bootstrap values greater than 75% were indicated. Black arrows indicate CV strains detected in this study and numerals indicate clades. Numbers of nucleotide substitutions are indicated by the scale bar.

**Figure 4 animals-13-01119-f004:**
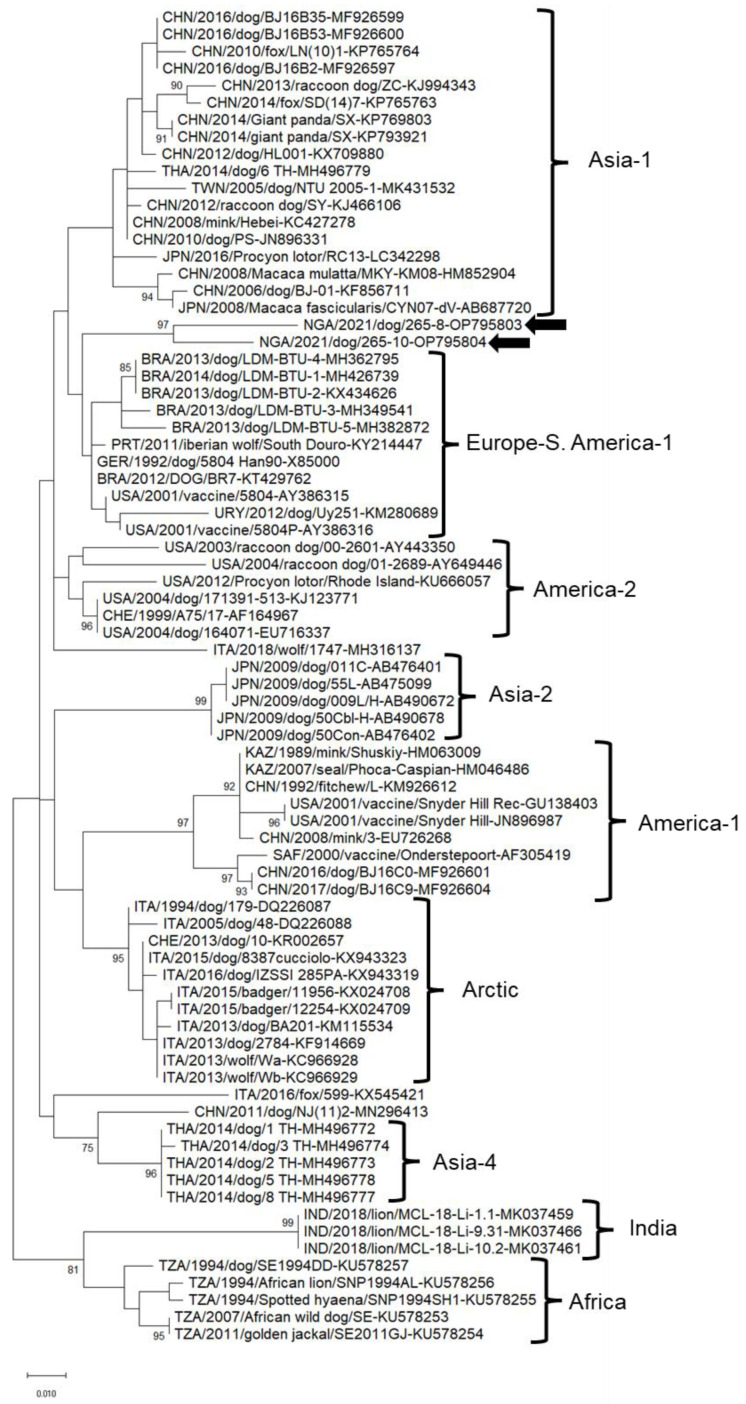
Phylogenetic tree based on the CDV partial nucleotide sequence of the hemagglutinin gene (267 nt). The maximum-likelihood method and Tamura-Nei model (four parameters) with a gamma distribution were used for the phylogeny. A total of 1000 bootstrap replicates were used to estimate the robustness of the individual nodes on the phylogenetic tree. Bootstrap values greater than 75% were indicated. Black arrows indicate CDV strains detected in this study. The numbers of nucleotide substitutions are indicated by the scale bar.

**Table 1 animals-13-01119-t001:** Screening results for Carnivore Protoparvovirus 1 (CPPV-1), canine circovirus (CanineCV), canine distemper virus (CDV), and canine adenovirus type 1 and 2 (CAdV-1/2).

Virus	Real-Time Results			CT Values		
	Pos	Neg	Total	Range	Mean	Median
CPPV-1	83	17	100	15 to 34	33	34
CanineCV	14	86	100	28 to 35	33	35
CDV	17	83	100	22 to 40	33	33
CAdV-1	0	100	100	-	-	-
CAdV-2	0	100	100	-	-	-

## Data Availability

The data presented in this study are available in this manuscript. Sequence data presented in this study are openly available in the GenBank database. Full VP2 sequences of CPV strains NGA/2021/265.21-4, NGA/2021/265.21-5, NGA/2021/265.21-11, NGA/2021/265.21-13, NGA/2021/265.21-18, NGA/2021/265.21-28, NGA/2021/265.21-42, NGA/2021/265.21-58, NGA/2021/265.21-59, NGA/2021/265.21-61, NGA/2021/265.21-71, NGA/2021/265.21-72, NGA/2021/265.21-76, NGA/2021/265.21-79, NGA/2021/265.21-82, NGA/2021/265.21-83, NGA/2021/265.21-85, NGA/2021/265.21-86, NGA/2021/265.21-88, NGA/2021/265.21-89, NGA/2021/265.21-98, and NGA/2021/265.21-99 were deposited in the GenBank under accession numbers. ON063543–ON063564, respectively. Partial rep gene sequences of CaCV strains NGA/2021/4, NGA/2021/5, NGA/2021/11, NGA/2021/48, NGA/2021/87, and NGA/2021/89 were submitted to the GenBank under accession numbers. OP654192–OP654197, respectively. Partial gene H and L sequences of CDV strains NGA/2021/265.21-8 and NGA/2021/265.21-10 were subjected to submission to the GenBank database under accession numbers OP795803 and OP795804, respectively.

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
