# Peer review of "Detection of Selected Canine Viruses in Nigerian Free-Ranging Dogs Traded for Meat Consumption"

_animals, 2023, doi:10.3390/ani13061119_

Round 1

Reviewer 1 Report

In this study free-ranging dogs traded for meat consumption in Nigeria were tested by molecular methods for some canine viruses: CPV, CAdV-1and2, CaCV and CDV. Dogs tested positive for CPV, CaCV and CDV, but not for CAdV-1and2. The genome of viruses identified were partially sequenced and phylogenetically analised. The study is well described and written, provides new data on the situation in Nigeria and deserves publication.

Minor revisions:

Abstract
Line 16: "Abstract" is repeated two times
Line 20 and elsewhere in the text: How many dogs corresponded to the 100 blood samples? 100 dogs or have some dogs been repeatedly sampled?

Materials and Methods
Subheading 2.4 GenBank Sequence Submission: it could be totally moved into the "Data Availability Statement"

Results (and consequently the Materials and Methods)
For CPV, CDV and CaCV, CT values have been reported but the detection limits of the assays used have not been reported in the MeM section. What was the detection limit of the assays adopted which allowed to consider positive or negative the samples tested?

Figure 4 - caption: For consistency with the captions of the other phylogenetic trees, I would suggest reporting the nucleotide length of the hemagglutinin gene sequence used for the analysis

Author Response

Reviewer 1

In this study free-ranging dogs traded for meat consumption in Nigeria were tested by molecular methods for some canine viruses: CPV, CAdV-1and 2, CaCV and CDV. Dogs tested positive for CPV, CaCV and CDV, but not for CAdV-1and2. The genome of viruses identified were partially sequenced and phylogenetically analised. The study is well described and written, provides new data on the situation in Nigeria and deserves publication.

General Reply to R1: We thank the referee for the appreciation of the manuscript. We appreciated the suggestions that contributed to improve the manuscript.

Minor revisions:

Abstract

R1.1 Line 16: "Abstract" is repeated two times

Reply to R1.1 We removed the second word “Abstract”

R1.2 Line 20 and elsewhere in the text: How many dogs corresponded to the 100 blood samples? 100 dogs or have some dogs been repeatedly sampled?

Reply to R1.2 A total of 100 blood samples were collected from 100 dogs. No dog has been sampled repeatedly in this study. As stated in the text: “Blood was collected by convenience sampling from 100 dogs slaughtered between August 2020 and March 2021, with an average of 3 samples collected weekly.”

Materials and Methods

R1.3 Subheading 2.4 GenBank Sequence Submission: it could be totally moved into the "Data Availability Statement"

Reply to R1.3 We agree with the referee. Accordingly, the sentence has been moved to the section "Data Availability Statement"

Results (and consequently the Materials and Methods)

R1.4 For CPV, CDV and CaCV, CT values have been reported but the detection limits of the assays used have not been reported in the MeM section. What was the detection limit of the assays adopted which allowed to consider positive or negative the samples tested?

Reply to R1.4 We agree with the referee that the information regarding the detection limits has not been reported in the manuscript. However, detection limits of the assays for the screening for CPV, CDV and CaCV have been indicated in the papers cited in the text.

As for CPV, in the paper Decaro et al., 2006 it was stated: The detection limits of the type-specific MGB probe assays were 101 and 102 DNA copies for types 2a and 2b, respectively (type 2a/2b assay), and 102 and 101 DNA copies for types 2b and 2c, respectively (type 2b/2c assay).”

As for CDV, in the paper Elia et al., 2006 it was stated: “The detection limit of the TaqMan assay was 1 × 102 and 3.13 × 102 copies for standard RNA and genomic RNA, respectively.”

As for CaCV, in the paper Li et al., 2013 it was stated:” Synthetic DNA fragments (≈150 bp) of the corresponding regions were used to produce a standard curve and an analytical sensitivity of 10 molecules.”

R1.5 Figure 4 - caption: For consistency with the captions of the other phylogenetic trees, I would suggest reporting the nucleotide length of the hemagglutinin gene sequence used for the analysis 

Reply to R1.5 We agree with the referee and we added this information in the figure caption of the CDV-based phylogenetic tree.

Reviewer 2 Report

Only minor comments: 

Page 4, line 163:  Extension at 15oC is incorrect.  

Page 5, line 166: Extencion at 30oC is incorrect.  

Figure 2:  Page 7, Line 250.  Complete the sentence "The black arrows indicate generated in this study".     indicate sequences?

Author Response

Reviewer 2

Only minor comments:

R2.1 Page 4, line 163:  Extension at 15°C is incorrect. 

Reply to R2.1 This was corrected

R2.2 Page 5, line 166: Extension at 30°C is incorrect. 

Reply to R2.2 This was corrected

R2.3 Figure 2:  Page 7, Line 250.  Complete the sentence "The black arrows indicate generated in this study".     indicate sequences?

Reply to R2.3 The word “strains” was added in the sentence. We thank the referee for the appropriate suggestion.

Reviewer 3 Report

Nice and interesting paper. I have enjoyed reading it. 

Author Response

Reviewer 3

Nice and interesting paper. I have enjoyed reading it.

General Reply to R3: We thank the referee for the appreciation of the manuscript. The efforts of the co-authors were repaid from these compliments.

Reviewer 4 Report

This report details the identification of canine viruses in dogs for meat production in Nigeria. The detection methods used in this study, as well as the resulting findings, are both convincing and supportive of the authors' conclusions. There are several points which should be addressed and revised prior to acceptance.

1. Line 75:

The UTRs is used once. No need to abbreviate.

2. Line 152:

The gene junction region between H and L genes is not called “intron”. The “gene junction region” or “intergenic region” is exact.

3. Line 212:

Why the 35 samples were selected? Please describe in detail.

4. Line 213:

What does the “single infections” means? This sentence should be modified to more clearly indicate the explanation.

5. Table 1:

The number of CDV Neg may be 83?

Author Response

Reviewer 4

This report details the identification of canine viruses in dogs for meat production in Nigeria. The detection methods used in this study, as well as the resulting findings, are both convincing and supportive of the authors' conclusions. There are several points which should be addressed and revised prior to acceptance.

General Reply to R4: We thank the referee for the appreciation of the manuscript. We appreciated the suggestions that contributed to improve the manuscript.

R4.1 Line 75: The UTRs is used once. No need to abbreviate.

Reply to R4.1 The acronym UTR was removed, as suggested.

R4.2 Line 152: The gene junction region between H and L genes is not called “intron”. The “gene junction region” or “intergenic region” is exact.

Reply to R4.2 The word “intron” was replaced by “intergenic region”, as suggested.

R4.3 Line 212: Why the 35 samples were selected? Please describe in detail.

Reply to R4.3 No selection was performed on the 83 samples tested positive to CPPV-1 by generic qPCR. Conversely, a total of 35 out of 83 CPPV-1 -positive samples could be characterized by minor groove binder (MGB) probe-based qPCR assays to differentiate CPV types 2a/2b and 2b/2c as well as vaccine CPV-2 strains from field variants.  

R4.4 Line 213: What does the “single infections” means? This sentence should be modified to more clearly indicate the explanation.

Reply to R4.4 The expression single infection was removed from the text, as suggested. Accordingly, we mentioned in the text that: “CPV-2c was identified in 77.1% (27/35) of the characterized samples and in co-infections with CPV-2a in 8.6% (3/35) of the samples.”

R4.5 Table 1: The number of CDV Neg may be 83?

Reply to R4.5 This was corrected. We thank the referee for the correction.